# TMPRSS2 promotes SARS-CoV-2 evasion from NCOA7-mediated restriction

Hataf Khan[1], Helena Winstone[1], Jose M. Jimenez-Guardeño[1], Carl Graham[1], Katie J. Doores[1], Caroline Goujon[2], David A. Matthews[3], Andrew D. Davidson[3], Suzannah J. Rihn[4], Massimo Palmarini[4], Stuart J. D. Neil[1], Michael H. Malim[1]*

1 Department of Infectious Diseases, School of Immunology and Microbial Sciences, King's College London, London, United Kingdom, 2 IRIM, CNRS, Université de Montpellier, Montpellier, France, 3 School of Cellular and Molecular Medicine, Faculty of Life Sciences, University Walk, University of Bristol, Bristol, United Kingdom, 4 MRC-University of Glasgow Centre for Virus Research (CVR), Glasgow, United Kingdom

* michael.malim@kcl.ac.uk

**Data Availability Statement:** All relevant data are within the manuscript and its Supporting Information files.

## Abstract

Interferons play a critical role in regulating host immune responses to SARS-CoV-2, but the interferon (IFN)-stimulated gene (ISG) effectors that inhibit SARS-CoV-2 are not well characterized. The IFN-inducible short isoform of human nuclear receptor coactivator 7 (NCOA7) inhibits endocytic virus entry, interacts with the vacuolar ATPase, and promotes endo-lysosomal vesicle acidification and lysosomal protease activity. Here, we used ectopic expression and gene knockout to demonstrate that NCOA7 inhibits infection by SARS-CoV-2 as well as by lentivirus particles pseudotyped with SARS-CoV-2 Spike in lung epithelial cells. Infection with the highly pathogenic, SARS-CoV-1 and MERS-CoV, or seasonal, HCoV-229E and HCoV-NL63, coronavirus Spike-pseudotyped viruses was also inhibited by NCOA7. Importantly, either overexpression of TMPRSS2, which promotes plasma membrane fusion versus endosomal fusion of SARS-CoV-2, or removal of Spike's polybasic furin cleavage site rendered SARS-CoV-2 less sensitive to NCOA7 restriction. Collectively, our data indicate that furin cleavage sensitizes SARS-CoV-2 Spike to the antiviral consequences of endosomal acidification by NCOA7, and suggest that the acquisition of furin cleavage may have favoured the co-option of cell surface TMPRSS proteases as a strategy to evade the suppressive effects of IFN-induced endo-lysosomal dysregulation on virus infection.

## Author summary

IFNs play a critical role in regulating host immune responses to virus infections. In cultured cell systems, SARS-CoV-2 can trigger robust IFN and innate immune responses through activation of cytoplasmic RNA sensors, and is also highly sensitive to inhibition by IFN treatment. Consistent with this, ISGs are induced in patients with COVID-19, and inborn errors in the IFN system are associated with severe COVID-19. Our understanding of the repertoire and mechanisms of action of the ISGs that impact SARS-CoV-2 is currently incomplete. In this study, we demonstrate that the IFN-inducible short isoform of

**Funding:** This work was funded by King's Together Rapid COVID-19 Call awards to MHM, SJDN and KJD; the Wellcome Trust (106223/Z/14/Z to MHM); a Huo Family Foundation Award to MHM, SJDN, and KJD; and NIAID Award U54 AI150472 to MHM. We acknowledge the Genotype-to-Phenotype UK National Virology Consortium funded by MRC/UKRI (MR/W005611/1) and Public Health England for providing viral isolates. HW and CG were supported by the MRC-KCL Doctoral Training Partnership in Biomedical Sciences (MR/N013700/1). This work was supported by the Department of Health via a National Institute for Health Research comprehensive Biomedical Research Centre award to Guy's and St. Thomas' NHS Foundation Trust in partnership with King's College London and King's College Hospital NHS Foundation Trust (MHM). The funders had no role in study design, data collection and analysis, decision to publish, or preparation of the manuscript.

**Competing interests:** The authors have declared that no competing interests exist.

NCOA7 is a potent inhibitor of SARS-CoV-2 infection. Our data support the presence of two distinct pathways of SARS-CoV-2 entry and reveal that only the pH-dependent endo-lysosomal pathway is efficiently inhibited by NCOA7. Unravelling the biological significance of the polybasic furin cleavage site in the viral Spike protein, and the concomitant use of TMPRSS2 for cell surface fusion is an area of intense research. Our findings suggest that the acquisition of the polybasic furin cleavage site in Spike may have driven the co-option of the cell surface TMPRSS2 protease to evade the antiviral effects of NCOA7-mediated perturbation of the endo-lysosomal system.

## Introduction

Coronaviruses (CoV) are enveloped, positive-sense RNA viruses with genomes of approximately 30 kb that have a broad host range and the capability to cross species barriers to infect humans [1,2]. Severe acute respiratory syndrome coronavirus 2 (SARS-CoV-2) is a betacoronavirus responsible for coronavirus disease-19 (COVID-19) [3,4]. It is the seventh coronavirus known to infect humans. SARS-CoV-1 and MERS-CoV are highly pathogenic whereas the seasonal coronaviruses, HCoV-HKU1, HCoV-NL63, HCoV-OC43 and HCoV-229E, circulate in the human population globally but typically cause mild symptoms [5–8].

The SARS-CoV-2 Spike protein initiates viral entry into host cells by interacting primarily with angiotensin-converting enzyme 2 (ACE2) on the host cell plasma membrane [9]. Additional factors such as neuropilin-1 (NRP1) and heparan sulphate can facilitate ACE2 dependent entry [10,11]. Recently, the Spike mutation E484D that is present in a number of variants of concern (VOCs) has been shown to allow entry in an ACE2 independent manner [12]. Spike is a trimeric type I transmembrane glycoprotein [13], and is organized into two main subunits, S1 and S2. S1 binds ACE2 and S2 contains the hydrophobic fusion peptide [13,14]. For entry into susceptible cells, SARS-CoV-2 Spike requires activation or priming by cellular protease-mediated cleavage at two distinct sites [15]. The first site is located at the S1/S2 junction and contains a polybasic motif Pro-Arg-Arg-Ala-Arg-Ser (P-R-R-A-R-S) [16]. The second cleavage site (S2´) is located within S2 immediately upstream of the fusion peptide and is responsible for triggering virus-cell membrane fusion [17]. Cell surface expressed TMPRSS2, a type II transmembrane serine protease, is thought to prime Spike at the S2´ site and promote virus entry at the plasma membrane in a pH-independent manner [18]. However, SARS-CoV-2 fusion can also occur in the endo-lysosomal system where the pH-dependent activity of various cysteine proteases known as cathepsins prime Spike at the S2´ site [19].

Viral glycoprotein polybasic sites can be key virulence determinants, as illustrated for highly pathogenic influenza viruses: for instance, the presence of the polybasic site in the hemagglutinin protein of H5N1 strains is associated with significantly higher virus titres in the respiratory tract [20–22]. Unlike SARS-CoV-1, the presence of a polybasic site in SARS-CoV-2 Spike is thought to be an important determinant of pandemic potential but its precise role in virus transmission and pathogenesis is unclear [23]. Furin, a proprotein convertase located in the Golgi that recognises and cleaves the sequence motif R-X-R/K-R where X can be any amino acid residue [24,25], has been shown to cleave the polybasic cleavage site in SARS-CoV-2 Spike in virus producing cells [16,18,26]. Structural studies have revealed that cleavage at the polybasic furin-cleavage site decreases the overall stability of SARS-CoV-2 Spike and facilitates the adoption of the open conformation that is required for Spike to bind the ACE2 receptor [27]. Consistent with this, deletion of the furin cleavage site in SARS-CoV-2 Spike promotes

the adoption of closed form of the Spike and prevents cell–cell as well as virus–cell fusion in some cell types [16,26,28,29].

IFNs are immunomodulatory cytokines produced in response to viral challenge following recognition of pathogen-associated molecular patterns (PAMPs) by host-encoded pattern recognition receptors (PRRs). They are pivotal to cell-intrinsic antiviral immunity, contributing to innate responses through the activation of ISGs. There are at least 1000 human ISGs, with the viral substrates and functionality of many remaining to be elucidated [30]. Previously, we demonstrated that the IFN-inducible 219-amino acid short isoform of human nuclear receptor coactivator 7 (NCOA7) is an inhibitor of viruses that enter cells via the endo-lysosomal pathway [31]. The long isoform of NCOA7 which is not IFN-inducible was originally identified as an oestrogen receptor-associated protein that localizes to the nucleus upon oestradiol treatment [32,33]. The short IFN-inducible isoform of NCOA7, studied herein, contains a unique amino-terminal region of 25 amino acids and shares the carboxy-terminal five exons of the long isoform [34]. The carboxy-terminal region of NCOA7 comprises a TLDc (TBC/LysM domain-containing) domain, the function of which is currently not fully understood [33]. IFN-inducible NCOA7 has been shown to interact with the vacuolar H+-ATPase (V-ATPase) and promotes endo-lysosomal vesicle acidification, lysosomal protease activity and the degradation of endocytosed antigen [31]. In analyzing influenza virus, NCOA7 function is manifested through the inhibition of fusion of the viral and endosomal membranes and the cytosolic entry of viral ribonucleoproteins [31].

Recent genome-wide CRISPR-Cas9 screens have identified V-ATPase subunits and cathepsin L as SARS-CoV-2 cofactors [35–37]. Given that NCOA7 binds the V-ATPase and promotes cathepsin activity, we sought to investigate whether NCOA7 regulates SARS-CoV-2 infection. We found that NCOA7 is a potent inhibitor of SARS-CoV-2 and other coronavirus infections. Consistent with its mechanism of action, overexpression of TMPRSS2, which potentiates plasma membrane fusion rather than endosomal fusion of SARS-CoV-2, attenuates NCOA7-mediated restriction of SARS-CoV-2. We further show that presence of the polybasic furin cleavage site in SARS-CoV-2 Spike sensitizes infection to the inhibitory consequences of aberrant endosomal acidification by NCOA7.

## Results

### IFN-inducible isoform of NCOA7 inhibits SARS-CoV-2 infection

To investigate the activity of NCOA7 against SARS-CoV-2 we used A549 cells, a lung epithelial cell line, stably expressing the SARS-CoV-2 receptor ACE2 (S1A Fig). A549-ACE2 cells were transduced with lentiviral vectors to express the IFN-inducible short isoform of NCOA7 or an unrelated control protein, E2crimson. To examine the effect of NCOA7 on infectious (wild-type) SARS-CoV-2, NCOA7 expressing or control A549-ACE2 cells were challenged with increasing doses of the SARS-CoV-2 PHE England 02/2020 strain. Virus replication was inhibited in NCOA7 expressing cells compared to the control at each viral dose, as measured by flow cytometric enumeration of nucleocapsid positive cells 48 h after infection (Figs 1A and S1B). Concordantly, the titre of SARS-CoV-2 produced by NCOA7 expressing cells was 8-fold lower (Fig 1B). and crystal violet staining of cell monolayers showed that NCOA7 reduced SARS-CoV-2 mediated cytopathic effects at lower doses of input virus (Fig 1C).

To investigate the effect of NCOA7 on Spike-mediated viral entry, NCOA7 or E2crimson expressing A549-ACE2 cells were challenged with 25, 50 or 100 μl of luciferase-expressing lentivirus vectors pseudotyped with SARS-CoV-2 Spike or amphotropic murine leukaemia virus (MLV) Envelope. Infection with the SARS-CoV-2 Spike pseudotype, as measured by luciferase production, was inhibited 4- to 5-fold in cells expressing NCOA7 compared to control cells at

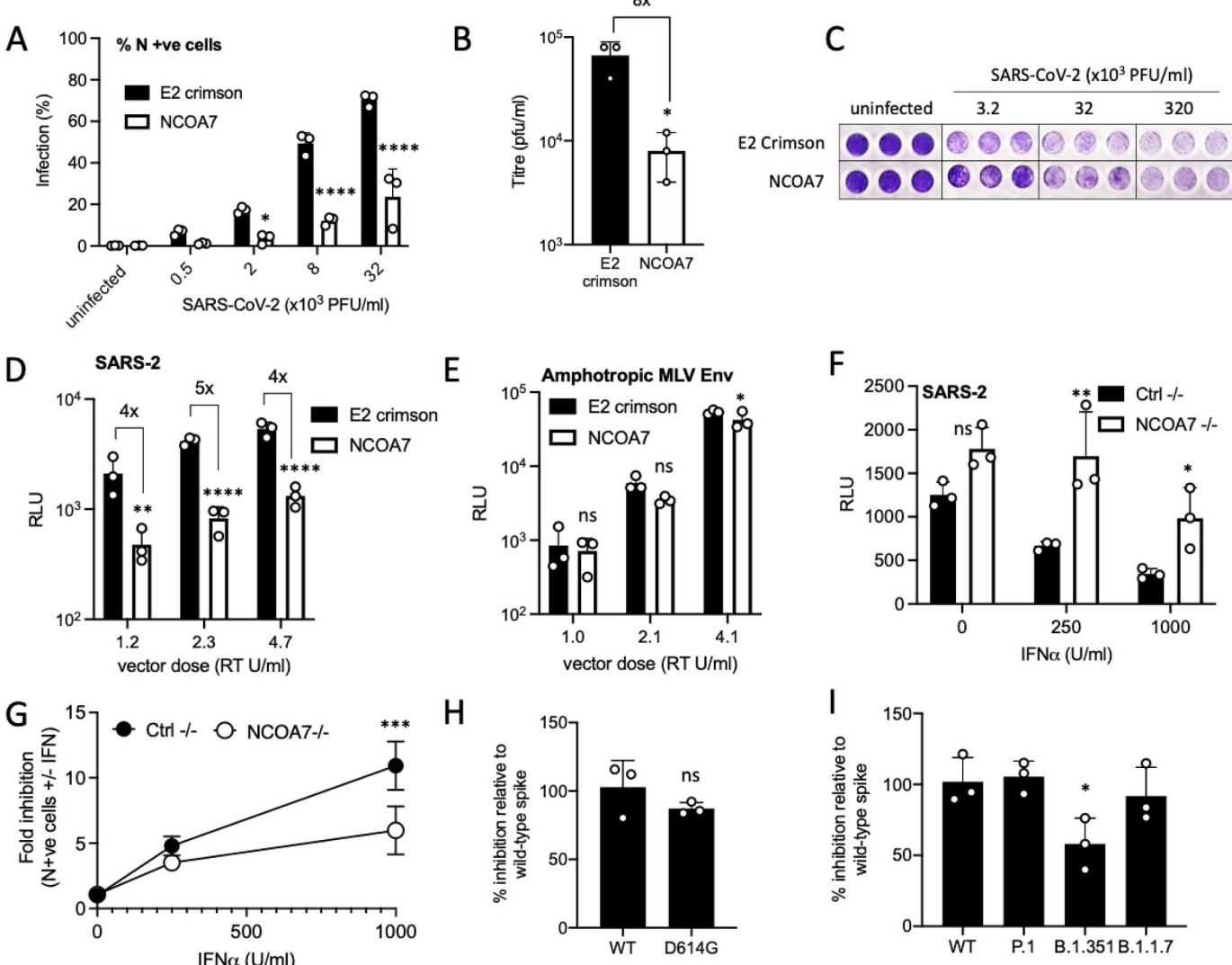

**Fig 1. NCOA7 inhibits SARS-CoV-2 infection.** (A) A549-ACE2 cells expressing E2crimson or NCOA7 were infected with replication-competent SARS-CoV-2 for 48 h. Nucleocapsid positive cells were enumerated by flow cytometry. (B) VeroE6 cells were incubated for 72 h with supernatants from A549-ACE2 cells expressing E2crimson or NCOA7 infected with replication-competent SARS-CoV-2 for 48 h. Plaque forming units were counted to calculate the titre. (C) Crystal violet staining of A549-ACE2 cells expressing E2crimson or NCOA7 after infection with replication competent SARS-CoV-2 for 72 h. (D) A549-ACE2 cells expressing E2crimson or NCOA7 were infected with SARS-CoV-2 Spike pseudotyped vector expressing luciferase. Luciferase activity was measured 48 h post-infection. (E) A549-ACE2 cells expressing E2crimson or NCOA7 were infected with Amphotropic MLV envelope pseudotyped vector expressing luciferase. Luciferase activity was measured 48 h post-infection. (F) A549-ACE2-Cas9 cells were transduced with lentiviral vectors expressing CRISPR non-targeting guide RNA (Ctrl-/-) or guide RNA targeting NCOA7. Treated with IFN-α for 24 h before challenge with SARS-CoV-2 Spike pseudotyped vector expressing luciferase. Luciferase activity was measured 48 h post-infection. (G) A549-ACE2-Cas9 cells were transduced with lentiviral vectors expressing CRISPR non-targeting guide RNA (Ctrl-/-) or guide RNA targeting NCOA7. Treated with IFN-α for 24 h before challenge with replication-competent SARS-CoV-2. Nucleocapsid positive cells were enumerated by flow cytometry 48 h post-infection. (H) A549-ACE2 cells expressing E2crimson or NCOA7 were infected with wild-type (WT) or D614G SARS-CoV-2 Spike pseudotyped vectors expressing luciferase. Luciferase activity was measured 48 h post-infection. (I) A549-ACE2 cells expressing E2crimson or NCOA7 were infected with wild-type (WT), P.1, B.1.351 or B.1.1.7 SARS-CoV-2 Spike pseudotyped vectors expressing luciferase. Luciferase activity was measured 48 h post-infection. For all the data n = 3, mean +/- SD. (A, D-G) analysed using two-way ANOVA. (B, H) analysed using unpaired t-test. (I) analysed using one-way ANOVA. * (p<0.05), ** (p<0.01), *** (p<0.001), **** (p<0.0001).

all doses tested (Fig 1D). In addition, expression of NCOA7 in a human bronchiolar epithelial cell line, BEAS-2B, which has been utilized as model for several infectious airway diseases also significantly inhibited Spike mediated viral entry (S1C Fig) [38–41]. As previously shown,

infection with the MLV pseudotype, where entry depends on macropinocytosis, was unaffected by NCOA7 expression (Fig 1E) [31,42]. To study the direct effect of NCOA7 mediated endosomal acidification on SARS-CoV-2 membrane fusion, we used a lipophilic dye-based fluorescence dequenching assay that uses SP-DiOC18. SARS-CoV-2 pseudotypes and infectious virus particles were labelled with SP-DiOC18. In the labelled particles, the green fluorescence of SP-DiOC18 is suppressed by self-quenching [43]. Fluorescence dequenching of SP-DiOC18 occurs following viral–endosome membrane fusion and there were significant decreases in the number of SP-DiOC18 positive cells in the presence of NCOA7 across a range of inocula, reflecting reduced membrane fusion (S1H and S1I Fig).

We complemented these results by assessing the contribution of endogenous NCOA7 to IFN-induced suppression of SARS-CoV-2 infection by using CRISPR-Cas9 genome editing to deplete A549-ACE2 cells of the IFN-inducible isoform of NCOA7. Control or NCOA7 knockout cells were treated with IFN-α for 24 h and then challenged with the SARS-CoV-2 Spike pseudotype or wild-type SARS-CoV-2. IFN-α effectively inhibited infection by both viruses in control cells, however, in NCOA7 knockout cells IFN-α was 2- to 3-fold less effective at inhibiting SARS-CoV-2 Spike pseudotype (Fig 1F) or wild-type SARS-CoV-2 (Fig 1G) infections, demonstrating that endogenous NCOA7 contributes to IFN-α mediated suppression of SARS-CoV-2.

Rapidly emerging SARS-CoV-2 VOCs are a major worry. One of the first variants to be identified contained the D614G mutation in Spike, which enhances viral infectivity and shifts Spike conformation towards an ACE2-binding and fusion competent state [44–46]. The B.1.1.7 variant emerged in the United Kingdom, the B.1.351 variant in South Africa and P.1 in Brazil. These newly emerged variants contain mutations and deletions in the receptor-binding domain (RBD) and N-terminal domain of the Spike protein, as well as in other proteins [47–52]. The D614G, P.1 and B1.351 Spike pseudotypes displayed enhanced infectivity relative to the wild-type Spike pseudotype in A549-ACE2 cells (S1D and S1E Fig) [44–46], whereas B.1.1.7 was indistinguishable (S1E Fig). To test the effects of NCOA7 on these variant Spikes, we infected NCOA7 expressing or control A549-ACE2 cells with pseudotypes bearing variant Spikes and observed that wild-type, D614G, B.1.1.7 and P.1 Spike pseudotypes were equally sensitive to NCOA7 (Figs 1H, 1I, S1F and S1G), whereas the B.1.351 Spike pseudotype was 50% less sensitive to NCOA7-mediated restriction (Figs 1I and S1G). Taken together, these data show that the IFN-inducible isoform of NCOA7 is an effective inhibitor of SARS-CoV-2 Spike function.

## NCOA7 inhibits pathogenic and seasonal coronavirus infections

Six other coronaviruses are known to infect humans. Seasonal coronaviruses include HCoV-229E, HCoV-OC43, HCoV-NL63, and HCoV-HKU1, which circulate in the human population globally and cause mild upper respiratory tract infections [8]. In contrast, infections with two others, SARS-CoV-1 and MERS-CoV, result in severe clinical presentations and substantial mortality [5,7]. HCoV-229E and HCoV-NL63 belong to the alphacoronavirus genus whereas HCoV-OC43, HCoV-HKU1, SARS-CoV-1 and MERS-CoV are betacoronaviruses [4]. Like SARS-CoV-2, SARS-CoV-1 and HCoV-NL63 use ACE2 as an entry receptor, HCoV-OC43 and HCoV-HKU1 use 9-O-acetylated sialic acids as a receptor [9,53–56], whereas MERS-CoV and HCoV-229E use dipeptidyl peptidase-4 (DPP4) and CD13 (also known as aminopeptidase N) as their receptors, respectively [57,58].

To test the activity of NCOA7 against seasonal and pathogenic human coronaviruses, NCOA7 expressing or control A549-ACE2 cells were infected with three doses of SARS-CoV-1 or HCoV-229E Spike pseudotypes. NCOA7 or E2crimson expressing U87-MG-ACE2 cells

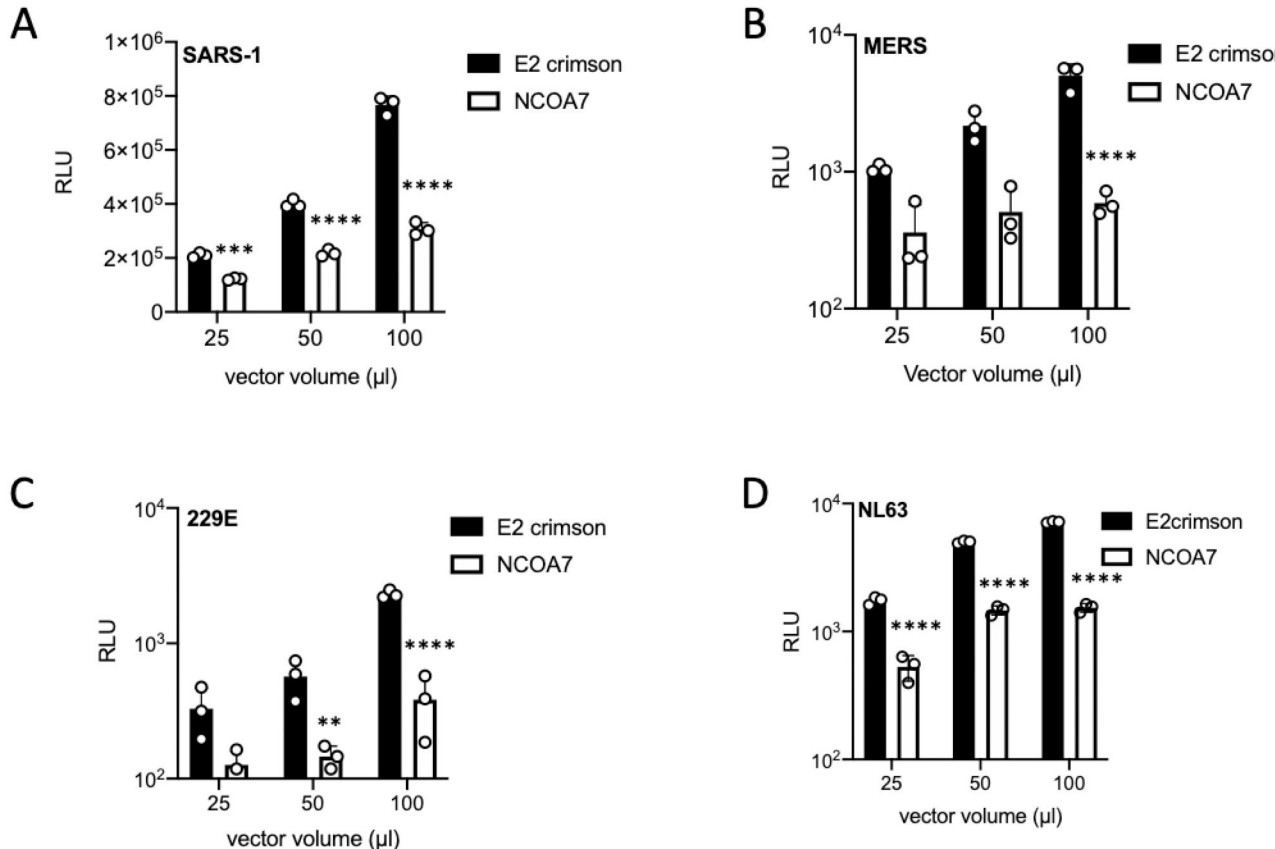

**Fig 2. NCOA7 inhibits pathogenic and seasonal coronavirus infections.** (A) A549-ACE2 cells expressing E2crimson or NCOA7 were infected with SARS-CoV-1 Spike pseudotyped vector expressing luciferase (SARS-1). Luciferase activity was measured 48 h post-infection. (B) U87-MG cells expressing E2crimson or NCOA7 were infected with MERS-CoV Spike pseudotyped vector expressing luciferase (MERS). Luciferase activity was measured 48 h post-infection. (C) A549-ACE2 cells expressing E2crimson or NCOA7 were infected with HCoV-229E Spike pseudotyped vector expressing luciferase (229E). Luciferase activity was measured 48 h post-infection. (D) U87-MG-ACE2 cells expressing E2crimson or NCOA7 were infected with HCoV-NL63 Spike pseudotyped vector expressing luciferase (NL63). Luciferase activity was measured 48 h post-infection. For all the data n = 3, mean +/- SD. All analysed using two-way ANOVA * (p<0.05), ** (p<0.01), *** (p<0.001), **** (p<0.0001).

were used to test the sensitivity of MERS-CoV and HCoV-NL63 Spike pseudotypes. Consistent with all of the HCoVs being able to enter cells via the endocytic pathway, we found that NCOA7 inhibited infection by SARS-CoV-1 (Fig 2A), MERS-CoV (Fig 2B), HCoV-229E (Fig 2C) and HCoV-NL63 (Fig 2D) Spike pseudotypes, demonstrating the pan-coronavirus antiviral activity of NCOA7.

## TMPRSS2 expression attenuates NCOA7-mediated restriction

As discussed earlier, SARS-CoVs can employ two routes for host cell entry that are determined by the localization of cellular proteases required for Spike protein activation. TMPRSS2, a type II transmembrane serine protease, has been shown to promote SARS-CoV-2 infection in a number of cell types [9], and is thought to prime Spike and allow virus-host membrane fusion at the cell surface in a pH-independent manner [59]. In TMPRSS2 expressing cells, SARS-CoVs entry can be blocked by the serine protease inhibitor camostat mesylate [9]. Given that NCOA7 inhibits virus entry through endosomes, we hypothesised that TMPRSS2 overexpression may promote virus fusion at the plasma membrane, and thereby bypass NCOA7-mediated restriction.

To test this, TMPRSS2 was stably expressed in NCOA7 expressing or control A549-ACE2 cells. As shown previously, TMPRSS2 expression enhanced infection by SARS-CoV-2 Spike pseudotypes by ~10-fold (Fig 3A). Consistent with this, infection with these pseudotypes was only inhibited by camostat mesylate in the TMPRSS2 expressing cells (Fig 3B). Next we examined the effect of TMPRSS2 on NCOA7-mediated inhibition of SARS-CoV-2 Spike pseudotype infection. In the absence of TMPRSS2, NCOA7 inhibited infection by ~8-fold (Fig 3C and 3D), however, in TMPRSS2 expressing cells the inhibitory effect dropped to ~3-fold (Fig 3C and 3D). Similarly, the inhibitory effect of NCOA7 on SARS-CoV-1 Spike pseudotyped vector infection was also suppressed by TMPRSS2 expression (Fig 3E and 3F). Our data demonstrate that TMPRSS2 overexpression attenuates NCOA7-mediated restriction likely by promoting virus fusion and entry at the plasma membrane.

## TMPRSS2 expression attenuates SARS-CoV-2 inhibition by endo-lysosomal pathway inhibitors

NCOA7 inhibits endosomal viral entry and interacts with the V-ATPase [31]. This interaction promotes endo-lysosomal acidification which in turn enhances the activity of proteases such as cathepsins [31]. Endosomal acidity or prolonged activation of cathepsins by NCOA7 may cause aberrant processing of Spike, functionally compromise other components of the virions and/or culminate in degradation in lysosomes. To support our observation that TMPRSS2 can rescue infection from the suppressive effects of NCOA7 mediated endosomal acidification, we pharmacologically manipulated endosomal acidification and endo-lysosomal proteases. Bafilomycin A1 (BafA1) is a specific inhibitor of V-ATPase whereas Niclosamide acts as a proton carrier which blocks endosomal acidification independently of V-ATPase [60,61]. E64d, however, is a broad inhibitor for cathepsin B, H, L, and calpain [62]. A549-ACE2 cells with or without TMPRSS2 were pre-treated with inhibitors for 2 h and then infected with SARS-CoV-2 Spike pseudotype. Like NCOA7 (Fig 4A), BafA1 (Fig 4B), E64d (Fig 4C) or Niclosamide (Fig 4D) treatment potently inhibited infection with Spike pseudotype in the absence of TMPRSS2, however, TMPRSS2 overexpression attenuated the inhibitory effects of each of these inhibitors. These results indicate that similar to the effect of TMPRSS2 on NCOA7 function, the SARS-CoV-2 inhibitory activities of pharmacological agents that manipulate endo-lysosomal acidification are also diminished by TMPRSS2.

## Furin cleavage site in Spike enhances NCOA7-mediated restriction

We noticed that infection with the SARS-CoV-1 Spike pseudotype was less sensitive to NCOA7 compared with the SARS-CoV-2 Spike pseudotype (compare Fig 3C and 3E). One of the most prominent differences between these Spike proteins is the presence of a polybasic furin cleavage site in SARS-CoV-2 [63]. Furin is a ubiquitously expressed calcium-dependent protease that is implicated in many viral infections via its cleavage of envelope glycoproteins, allowing not only virus-cell membrane fusion but also cell-to-cell fusion and syncytium formation [64]. Furin localises to the trans-Golgi network and acts on Spike during viral production [64]. In the context of SARS-CoV-2, the presence of a polybasic site in Spike is thought to be a determinant of pandemic potential, though its precise role(s) in virus infection and transmission remains unclear [23].

To test whether furin cleavage of Spike influences NCOA7-mediated restriction, we deleted the furin cleavage site (ΔPRRA) in SARS-CoV-2 Spike and added a furin cleavage site (PRRA) to SARS-CoV-1 Spike. NCOA7 expressing or control A549-ACE2 cells were infected with pseudotypes bearing wild-type or ΔPRRA SARS-CoV-2 Spike. NCOA7 inhibited wild-type virus infection by ~6-fold (Fig 5A and 5C), whereas infection with ΔPRRA Spike bearing virus

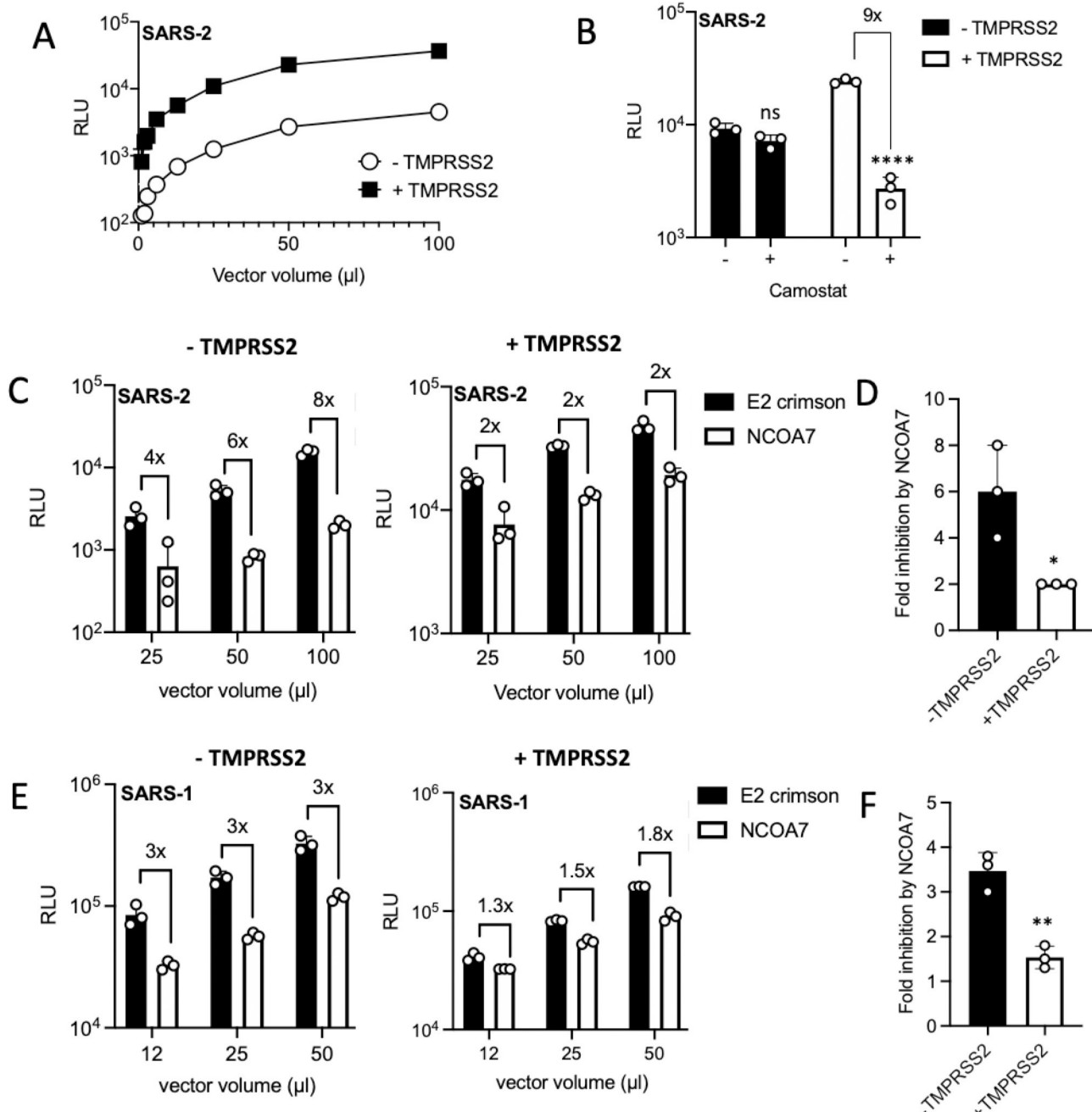

**Fig 3. TMPRSS2 overexpression attenuates NCOA7-mediated restriction.** (A) A549-ACE2 cells with or without TMPRSS2 were infected with increasing volumes of SARS-CoV-2 Spike pseudotyped vector expressing luciferase (SARS-2). Luciferase activity was measured 48 h post-infection. (B) A549-ACE2 cells with or without TMPRSS2 were treated with 100 µM camostat for 2 h and then infected with SARS-CoV-2 Spike pseudotyped vector expressing luciferase. Luciferase activity was measured 48 h post-infection. (C) A549-ACE2 cells expressing E2crimson or NCOA7 with (right) or without TMPRSS2 (left) were infected with SARS-CoV-2 Spike pseudotyped vector expressing luciferase. Luciferase activity was measured 48 h post-infection. (D) Fold inhibition of SARS-CoV-2 Spike pseudotyped vector infection by NCOA7 in the presence or absence of TMPRSS2 from Fig 3C. (E) A549-ACE2 cells expressing E2crimson or NCOA7 with (right) or without TMPRSS2 (left) were infected with SARS-CoV-1 Spike pseudotyped vector expressing luciferase (SARS-1). Luciferase activity was measured 48 h post-infection. (F) Fold inhibition of SARS-CoV-1 Spike pseudotyped vector infection by NCOA7 in the presence or absence of TMPRSS2 from Fig 3E. For all the data n = 3, mean +/- SD. (B) analysed using two-way ANOVA. (D, F) analysed using unpaired t-test. * ($p<0.05$), ** ($p<0.01$), *** ($p<0.001$), **** ($p<0.0001$).

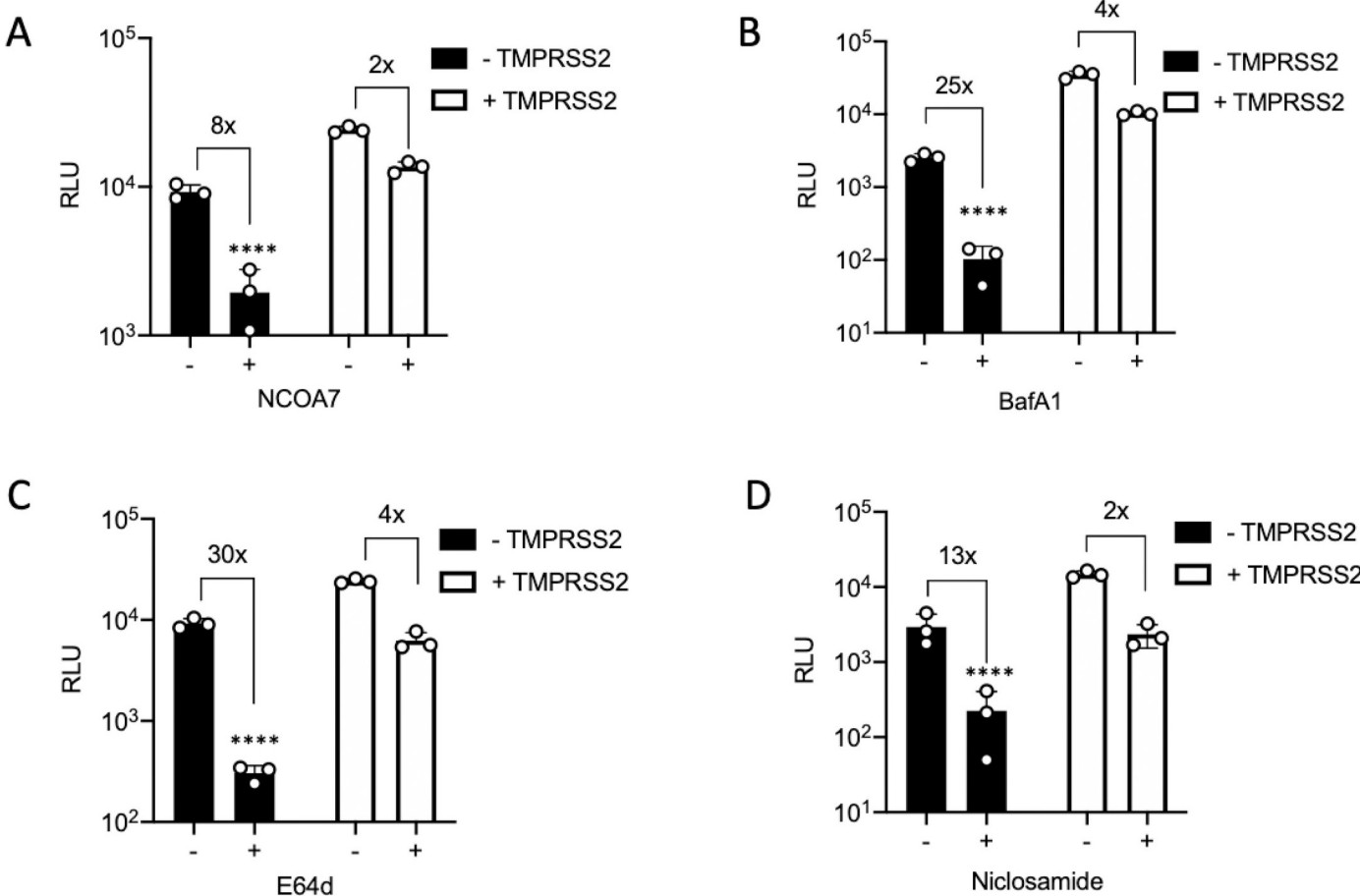

**Fig 4. TMPRSS2 attenuates SARS-CoV-2 inhibition by endo-lysosomal pathway inhibitors.** (A) A549-ACE2 cells with or without NCOA7 and TMPRSS2 were infected with SARS-CoV-2 Spike pseudotyped vector expressing luciferase. Luciferase activity was measured 48 h post-infection. (B) A549-ACE2 cells with or without TMPRSS2 were treated with 100 nM BafilomycinA1 (BafA1: V-ATPase inhibitor) or left untreated for 2 h and then infected with SARS-CoV-2 Spike pseudotyped vector expressing luciferase. Luciferase activity was measured 48 h post-infection. (C) A549-ACE2 cells with or without TMPRSS2 were treated with 10 μM E64d (cathepsin inhibitor) or left untreated for 2 h and then infected with SARS-CoV-2 Spike pseudotyped vector expressing luciferase. Luciferase activity was measured 48 h post-infection. (D) A549-ACE2 cells with or without TMPRSS2 were treated with 5 μM Niclosamide (neutralise pH) or left untreated for 2 h and then infected with SARS-CoV-2 Spike pseudotyped vector expressing luciferase. Luciferase activity was measured 48 h post-infection. For all the data n = 3, mean +/- SD. All analysed using two-way ANOVA. $^*$ ($p<0.05$), $^{**}$ ($p<0.01$), $^{***}$ ($p<0.001$), $^{****}$ ($p<0.0001$).

was inhibited by ~3-fold (Fig 5B and 5C), an ~50% decrease in NCOA7 sensitivity. Conversely, infections with pseudotypes bearing wild-type (Fig 5D) or PRRA containing (Fig 5E) SARS-CoV-1 Spike revealed that addition of the polybasic furin cleavage site enhanced the inhibitory effect of NCOA7 from 2- to 4-fold (Fig 5F). Finally, to test whether the polybasic furin cleavage site has a similar effect in the context of replication-competent SARS-CoV-2 infection, control or NCOA7 expressing A549-ACE2 cells were infected with infectious wild-type or ΔPRRA SARS-CoV-2 (Fig 5G). Consistent with the results with pseudotypes, the ΔPRRA virus was inhibited ~50% less effectively by NCOA7 when compared to wild-type SARS-CoV-2 (6-fold to 3-fold) (Fig 5G and 5H). Our data therefore suggest that furin cleavage renders SARS-CoV-2 Spike more prone to the detrimental effects of endo-lysosomal acidification induced by NCOA7.

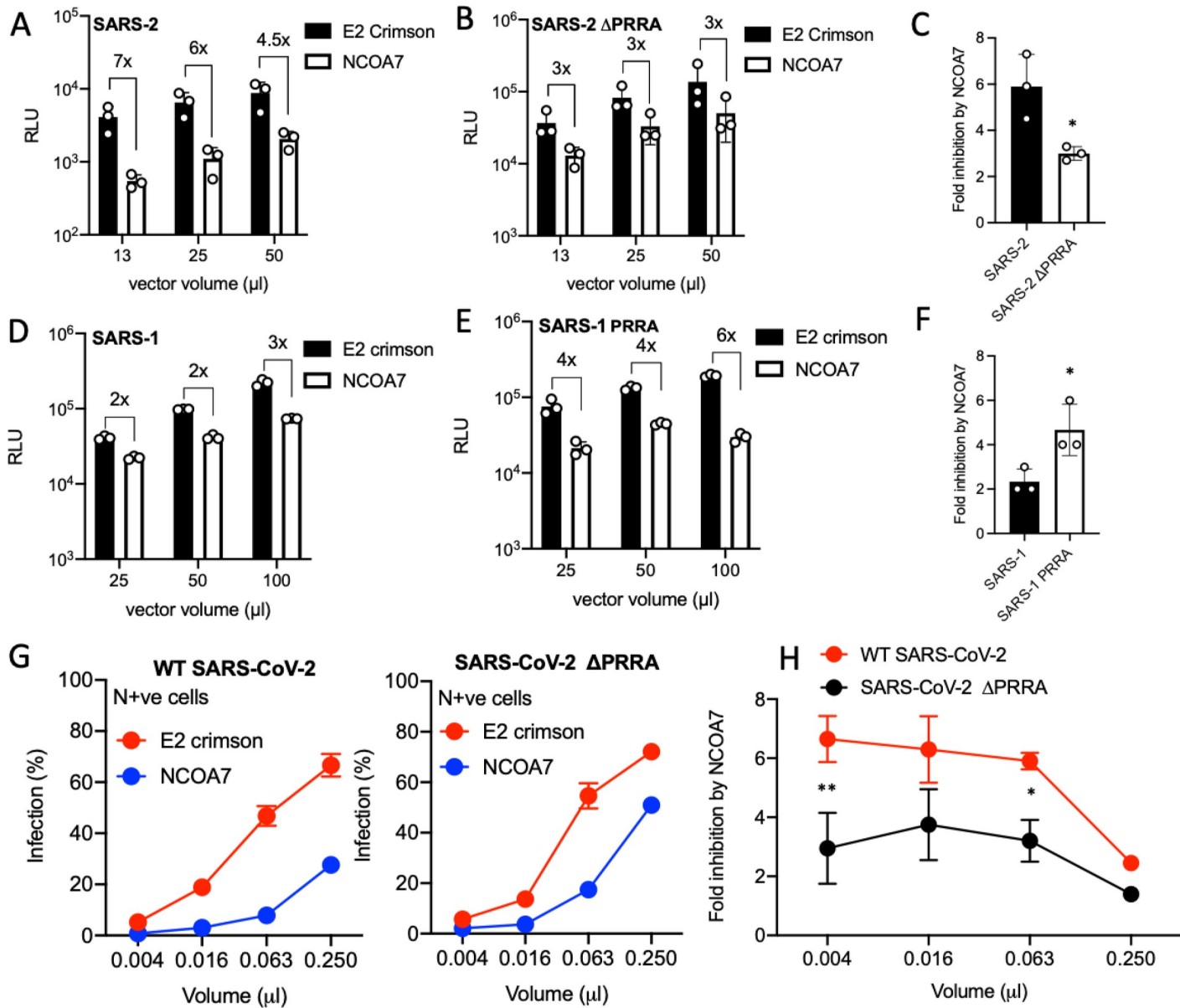

**Fig 5. Furin cleavage site in Spike enhances NCOA7-mediated restriction.** (A) A549-ACE2 cells expressing E2crimson or NCOA7 were infected with SARS-CoV-2 Spike pseudotyped vector expressing luciferase. Luciferase activity was measured 48 h post-infection. (B) A549-ACE2 cells expressing E2crimson or NCOA7 were infected with SARS-CoV-2ΔPRRA Spike pseudotyped vector expressing luciferase. Luciferase activity was measured 48 h post-infection. (C) Fold inhibition of SARS-CoV-2 WT or ΔPRRA Spike pseudotyped vector infection by NCOA7 from Fig 5A. (D) A549-ACE2 cells expressing E2crimson or NCOA7 were infected with SARS-CoV-1 Spike pseudotyped vector expressing luciferase. Luciferase activity was measured 48 h post-infection. (E) A549-ACE2 cells expressing E2crimson or NCOA7 were infected with SARS-CoV-1 PRRA Spike pseudotyped vector expressing luciferase. Luciferase activity was measured 48 h post-infection. (F) Fold inhibition of SARS-CoV-1 WT or PRRA Spike pseudotyped vector infection by NCOA7 from Fig 5E. (G) A549-ACE2 cells expressing E2crimson or NCOA7 were infected with increasing volumes of replication competent wild-type (WT) SARS-CoV-2 (left) or SARS-CoV-2 ΔPRRA (right) for 48 h. Nucleocapsid positive cells were enumerated by flow cytometry. (H) Fold inhibition of wild-type SARS-CoV-2 or SARS-CoV-2 ΔPRRA infection by NCOA7 calculated from Fig 5E. For all the data n = 3, mean +/- SD. (F) analysed using two-way ANOVA. (C, F) analysed using unpaired t-test. * (p<0.05), ** (p<0.01), *** (p<0.001), **** (p<0.0001).

## Discussion

IFNs play a critical role in regulating host immune responses to SARS-CoV-2 [65,66]. SARS-CoV-2 infection can trigger a robust innate immune response *in vitro* through activation of the cytoplasmic RNA sensors, RIG-I and MDA5 [67,68] and is highly sensitive to inhibition by

IFN treatment, upregulation of ISGs has been observed in COVID-19 patient samples, and inborn errors in the IFN system are associated with severe COVID-19 [69–74]. The identity and mechanisms of action of many ISGs that impact SARS-CoV-2 are still elusive. Recent work has identified *LY6E*, *AXIN2*, *CH25H*, *EPSTI1*, *GBP5*, *IFIH1*, *IFITM2*, and *IFITM3* as negative regulators of SARS-CoV-2 [75–79]. Furthermore, genetic screens have identified V-ATPase subunits, cathepsin L and other endosomal proteins, such as RAB7, as SARS-CoV-2 cofactors [35–37]. NCOA7 interacts with V-ATPase, and promote endo-lysosomal acidification and lysosomal protease activity [31]. In the present study, we focused on defining the activity of the IFN-inducible short isoform of NCOA7 against SARS-CoV-2. We used ectopic expression (Fig 1A–1D) and gene knockout (Fig 1F and 1G) to demonstrate that NCOA7 inhibits SARS-CoV-2 Spike pseudotyped virus and replication-competent virus infections in lung epithelial cells. We also show that entry of other pathogenic coronaviruses, SARS-CoV-1 (Fig 2A) and MERS-CoV (Fig 2B), and the seasonal coronaviruses, HCoV-229E (Fig 2C) and HCoV-NL63 (Fig 2C), is inhibited by NCOA7, highlighting the breadth of NCOA7's anti-coronavirus activity.

The human airway epithelium is an important site of SARS-CoV-2 infection and replication [9,80]. The virus can then disseminate to other tissues and cause multi-organ dysfunction in COVID-19 patients [81]. Viral fusion requires proteolytic cleavage of the Spike by cellular proteases, furin, cathepsins and TMPRSS2 [9,82–84]. The complementarity and interchangeability of these proteases, in allowing SARS-CoV-2 entry into cells might explain the wider tropism and transmission characteristics of this virus compared to SARS-CoV-1. Our data support the presence of two distinct pathways of SARS-CoV-2 entry, and we reveal that the pH-dependent endo-lysosomal pathway is efficiently inhibited by NCOA7. The pH-independent pathway is thought to be important for pathogenesis and virus propagation [85]. This is supported by the fact that the polybasic furin cleavage site is conserved in SARS-CoV-2 isolated from COVID-19 patients even though it can be selected against during *in vitro* passage [63,86].

We have determined that SARS-CoV-2 is more sensitive to NCOA7 than SARS-CoV-1 (Fig 3C and 3E). The most obvious difference between the SARS-CoV-1 and -2 Spike proteins is the presence of the polybasic furin cleavage site at S1/S2 junction, which is thought to be an important determinant of pandemic potential [16,63,85]. Thus, SARS-CoV-2 Spike, but not SARS-CoV-1, is cleaved at this site in the virus-producing cells [79], and it is notable that this feature has not previously been observed in many SARS-like coronaviruses identified in bats [87–89]. We found that loss of the furin cleavage site decreased the sensitivity of SARS-CoV-2 to NCOA7 (Fig 5A–5C); conversely, addition of the furin cleavage site to SARS-CoV-1 Spike enhanced the inhibitory effect of NCOA7 against SARS-CoV-1 (Fig 5A–5C). Together, our data indicate that acquisition of the polybasic furin cleavage site in SARS-CoV-2 sensitises its Spike to the detrimental consequences of NCOA7-mediated endosomal acidification in cells expressing low levels of TMPRSS2, possibly owing to the relative instability of SARS-CoV-2 Spike compared to the Spike proteins of SARS-CoV-1 and bat RaTG13 [19,27]. This is consistent with other studies showing that SARS-CoV-2 utilizes the endocytic entry pathway less efficiently than SARS-CoV-1 in the absence of TMPRSS2, thus correlating furin-cleavage during virus production with greater dependence on TMPRSS2 for efficient entry at the plasma membrane and lower reliance on cathepsins [59]. Similarly, recent studies reveal that the polybasic site in Spike facilitates evasion from IFN-induced transmembrane protein (IFITM)-mediated restriction and is required for transmission [85], and that activation of Spike by TMPRSS2 at the cell surface allows SARS-CoV-2 to evade inhibition by IFITM proteins that are present in endosomal compartments [79,85]. Together, these findings suggest that the presence of the polybasic furin cleavage site in SARS-CoV-2 Spike may have driven the co-option of cell

surface TMPRSS enzymes to evade the detrimental effects of endosomal restriction factors as well as endo-lysosomal acidification (including its amplification by factors such as NCOA7) on viral infection.

Finally, a recent study by Ghosh et al. (2020) showed that betacoronaviruses use lysosomes for egress instead of the biosynthetic secretory pathway [90], raising the possibility of a role for NCOA7 during late stages of virus life cycle. Importantly, it was also reported that lysosomes are deacidified, and proteolytic enzymes are inactive in infected cells, potentially indicating antagonism of NCOA7. In other work, mass spectrometry analyses of SARS-CoV-2 proteins have identified interactions between V-ATPase subunits and viral Matrix (M) and non-structural protein 6 (Nsp6) proteins [91]. Accordingly, there are a number of emerging questions centred on the dynamic interplay between NCOA7 and SARS-CoV-2: answering these may inform the strategic development of therapeutic interventions.

## Methods and materials

### Plasmids

NCOA7 variant 6 (NM_001199622.1) (encoding isoform 4 NP_001186551.1) or E2crimson cDNAs were inserted into pRRL.sin.cPPT.SFFV/IRES-puro.WPRE as described before [31]. Codon-optimised SARS-CoV-1, MERS-CoV, HCoV-229E and HCoV-NL63 Spike genes were synthesised in pcDNA3.1 from GeneArt. Codon-optimised SARS-CoV-2 Spike in pcDNA3.1 and ACE2 expression constructs were kindly provided by Dr Nigel Temperton [92]. SARS-CoV-2 Spike P.1, B.1.1.7, B1.351 were synthesised in pcDNA3.1 from GENEWIZ. The ACE2 cDNA was inserted into pMIGR1-blast using NotI and XhoI to create pMIGR1-blast-ACE2. TMPRSS2 lentiviral expression construct, RRL.sin.cPPT.SFFV/TMPRSS2(variant1).IRES-neo. WPRE, has been described before [31]. The LentiCas9-Blast vector was a gift from F. Zhang (Addgene). The guide RNA coding sequences used were as follows: g1-CTRL 5′-agcacg-taatgtccgtggat, g1-NCOA7 5′-aaaaggctcttcgtcaggtc. The NCOA7 guide was designed to target the first coding exon of NCOA7 variant 6, which is unique to the short IFN-inducible isoform 4 studied herein. The RNA guides were cloned into LentiGuide-Neo vector as described [31]. SARS-CoV-1 and SARS-CoV-2 Spike mutants were generated with Q5 Site-Directed Mutagenesis Kit (E0554) following the manufacturer's instructions using the following primers as described in [79]:

SARS-CoV-2 ΔPRRA Spike (Fwd: AGAAGCGTGGCCAGCCAG, Rev: GCTATTG GTCTGGGTCTGGTAG)

SARS-CoV-1 PRRA Spike (Fwd: AGAGCCCGGAGCACCAGCCAGAAA, Rev: TCTA GGCAGCAGAGACACGGTGTG)

### Cell lines

All cell lines were cultured in DMEM (Gibco) with 10% fetal bovine serum (FBS, Invitrogen) plus 100 U/ml penicillin and 100 μg/ml streptomycin (Pen/Strep; Gibco), and incubated at 37˚C, 5% $CO_2$. A549 and U87MG cells stably expressing ACE2 were generated by transduction with pMIGR1-blast-ACE2 vector and maintained with 10 μg/ml blasticidin selection. A549-ACE2 and U87MG-ACE2 cells stably expressing NCOA7 or E2crimson were generated by transduction with RRL.sin.cPPT.SFFV/IRES-puro.WPRE-containing vectors expressing NCOA7 or E2crimson and maintained with 2 μg/ml puromycin and 10 μg/ml balsticidin. A549-ACE2-NCOA7/E2crimson cells stably expressing TMPRSS2 were generated by transduction with RRL.sin.cPPT.SFFV/TMPRSS2(variant1).IRES-neo.WPRE-containing vector and maintained with 2 μg/ml puromycin, 10 μg/ml blasticidin and 800 μg/ml neomycin. For CRISPR–Cas9-mediated gene disruption, A549-ACE2 cells stably expressing Cas9 were first

generated by transduction with the LentiCas9-Blast vector followed by blasticidin selection at 10 μg/ml. Cas9-expressing A549-ACE2 cells were then transduced with guide RNA expressing Lentiguide-Neo vectors and cells selected with antibiotics for at least 15 days. When indicated, IFN-α (INTRON A, Merck, Sharpe & Dohme Corp.) was added for 24 h before virus infection challenge. Cells were treated with 100 nM bafilomycin A1 (Sigma), 10 μM E64d (Sigma) or 100 μM Camostat mesylate (Sigma) for 2 h before virus challenge.

## Pseudotyped lentiviral vector production and infection

Pseudotyped lentiviral vectors were produced in a 10-cm dish seeded the day before with HEK293T cells. Cells were transfected using 20 μl of TransIT-2020 (Mirus Bio) with: 4 μg of pCSLW [93], 4 μg of p8.91 [94] and 2 μg of the relevant glycoprotein expression plasmid. The medium was changed 24 h post-transfection and the supernatant was harvested 48 h post-transfection. Supernatant was filtered through a 0.45-μm filter (Sartorious) and stored at −80˚C until required. Viral stocks were quantified by measuring reverse transcriptase activity by qPCR using a SYBR Green-based product-enhanced PCR assay (SG-PERT) as described [95]. Viral supernatant was then typically used at ~eight dilutions to transduce each cell line of interest for 48 h and luciferase activity was measured with Promega Steady-Glo kit (E2550) using a Victor X3 Multilabel Reader (Perkin Elmer). Typically, data are presented from three doses that were evidently in the quantitative range.

## Replication competent SARS-CoV-2

SARS-CoV-2 Strain England/02/2020 was obtained from Public Health England. SARS-CoV-2 strain England/02/2020 in which the RRAR furin cleavage site had been deleted has been described [10,96]. The viruses were propagated by infecting 60–70% confluent Vero-E6 cells in T75 flasks. Supernatant was harvested 72 h post-infection after visible cytopathic effect, and filtered through a 0.22-μm filter (Sartorius) to eliminate debris, aliquoted and stored at −80˚C. The infectious virus titre was determined by plaque assay in Vero-E6 cells. All work with infectious SARS-CoV-2 was carried out in a Containment Level 3 facility (Health and Safety Executive approvals, CBA1.295.20.1 and GM386/20.2).

## Immunoblotting

Cells were lysed in buffer containing 50 mM Tris pH 8, 150 mM NaCl, 1 mM EDTA, 10% (v/v) glycerol, 1% (v/v) Triton X100, 0.05% (v/v) NP40 supplemented with protease inhibitors (Roche) and boiled in 6X Laemmli buffer for 10 min. Proteins were separated by SDS-PAGE on 12% polyacrylamide gels. Proteins were transferred to a Hybond ECL membrane (Amersham biosciences) using a semi-dry transfer system (Biorad) and probed with primary antibodies: 1:1000 rabbit anti-ACE2 (Abcam, Ab108209) or 1:10000 mouse anti-tubulin (Sigma, T5168) and secondary antibodies: anti-mouse IgG IRdye 800CW (LI-COR Biosciences) or anti-rabbit IgG IRdye 800CW (LI-COR Biosciences).

## Intracellular SARS-CoV-2 nucleocapsid staining

Cells were resuspended 48 h post-infection and fixed in 5% formaldehyde in PBS (Gibco) for 30 minutes at room temperature. Cells were permeabilised with 0.1% Triton X100 (Sigma) in PBS for 10 min. Cells were blocked with 5% FCS in PBS (blocking buffer) for 30 min, and incubated with sheep anti-SARS-CoV-2 nucleocapsid antibody (1:10000) [97] in blocking buffer for 1 h. Samples were incubated with anti-sheep AlexaFluor 488 IgG (1:500, Invitrogen) in blocking buffer for 1 h. Cells were washed twice with PBS, resuspended in PBS, and

nucleocapsid positive cells were enumerated by flow cytometry using a FACSCanto II (BD Biosciences). Data were analysed with Graphpad Prism 9.

## Membrane fusion assay using SP-DiOC18

SARS-CoV-2 particles were labelled with the self-quenching dye SP-DiOC18 (Invitrogen). SP-DiOC18 was added to 1 ml of virus stock containing approximately $2.5 \times 10^7$ PFU/ml at a final concentration of 0.2 μM. The tube was protected from light, incubated while rolling for 1 h at room temperature and subsequently stored at −80˚C or used as required. Cells were incubated for 1.5 h with the labelled virus, harvested with trypsin, fixed with 4% PFA for 30 min and analysed by FACSCanto II (BD Biosciences) using the 488 nm laser for excitation.

## Crystal violet staining

Cells were fixed in 5% formaldehyde in PBS for 30 min at room temperature after 72 h infection. Cells were then stained with crystal violet (Pro-Lab Diagnostics) solution for 5 min. Cell were washed with PBS and then pictures were taken.

## Supporting information

**S1 Fig. NCOA7 inhibits SARS-CoV2 infection.** (A) Immunoblot showing ACE2 expression in A549 cells after transduction with ACE2 expressing vector. Tubulin is included as a loading control. (B) Gating strategy for flow cytometry analysis of SARS-CoV-2 nucleocapsid (N) positive cells. Cells were distinguished from debris using the forward scatter (FCS, related to the cell size) and side scatter (SSC, related to the cell granularity). Selected population was gated for N positive cells (FITC channel) using uninfected cells as a negative control. SARS-CoV-2 infected cells were then gated for N. (C) BEAS-2B-ACE2 cells expressing E2crimson or NCOA7 were infected with SARS-CoV-2 Spike pseudotyped vector expressing luciferase. Luciferase activity was measured 48 h post-infection. (D) A549-ACE2 cells were infected with increasing volumes of wild-type (WT) or D614G SARS-CoV-2 Spike pseudotyped vectors expressing luciferase. Luciferase activity was measured 48 h post-infection. (E) A549-ACE2 cells were infected with increasing volumes of wild-type (WT), P.1, B.1.351 or B.1.1.7 SARS-CoV-2 Spike pseudotyped vectors expressing luciferase. Luciferase activity was measured 48 h post-infection. (F) A549-ACE2 cells expressing E2crimson or NCOA7 were infected with wild-type (WT) or D614G SARS-CoV-2 Spike pseudotyped vectors expressing luciferase. Luciferase activity was measured 48 h post-infection. (G) A549-ACE2 cells expressing E2crimson or NCOA7 were infected with wild-type (WT), P.1, B.1.351 or B.1.1.7 SARS-CoV-2 Spike pseudotyped vectors expressing luciferase. Luciferase activity was measured 48 h post-infection. (H) A549-ACE2 cells expressing E2crimson or NCOA7 were infected with SP-DiOC18 labelled SARS-CoV-2 for 1.5 h. SP-DiOC18 positive cells were enumerated by flow cytometry. (I) A549-ACE2 cells expressing E2crimson or NCOA7 were infected with SP-DiOC18 labelled SARS-CoV-2 Spike pseudotyped vector for 1.5 h. SP-DiOC18 positive cells were enumerated by flow cytometry. (C, H, I) analysed using two-way ANOVA. * ($p<0.05$), ** ($p<0.01$), *** ($p<0.001$), **** ($p<0.0001$). (TIF)

## Acknowledgments

For the purpose of open access, the author has applied a CC BY public copyright license to any Author Accepted Manuscript version arising from this submission.

## Author Contributions

**Conceptualization:** Hataf Khan, Stuart J. D. Neil, Michael H. Malim.

**Data curation:** Hataf Khan.

**Formal analysis:** Hataf Khan, Michael H. Malim.

**Funding acquisition:** Michael H. Malim.

**Investigation:** Hataf Khan.

**Methodology:** Jose M. Jimenez-Guardeño.

**Project administration:** Michael H. Malim.

**Resources:** Helena Winstone, Jose M. Jimenez-Guardeño, Carl Graham, Katie J. Doores, Caroline Goujon, David A. Matthews, Andrew D. Davidson, Suzannah J. Rihn, Massimo Palmarini, Stuart J. D. Neil.

**Supervision:** Michael H. Malim.

**Writing – original draft:** Hataf Khan.

**Writing – review & editing:** Hataf Khan, Stuart J. D. Neil, Michael H. Malim.

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
