## [Decision Letter · Decision Letter 0]

5 Aug 2021

Dear Prof. Malim,

Thank you very much for submitting your manuscript "TMPRSS2 promotes SARS-CoV-2 evasion from NCOA7-mediated restriction" for consideration at PLOS Pathogens. As with all papers reviewed by the journal, your manuscript was reviewed by members of the editorial board and by several independent reviewers. In light of the reviews (below this email), we would like to invite the resubmission of a significantly-revised version that takes into account the reviewers' comments.

Please pay particular attention to central themes across the several reviews:

(1) The lack of any statistics  was noted by all four reviewers and must be rectified this age of rigor and reproducibility. For that matter, I agree that a more quantifiable way of measuring virus input is required for cross-comparison across experiments and figures (Reviewer 2).

(2) At the the end of the day, it is still unclear how the interplay between furin cleavage site and TMPRSS2 activity  modulates NCOA7 restriction of SARS2 infection (Reviewers1 and 4). Sone of the data provided appear counter-intuitive. Instead of speculation, I urged the authors to provide data that directly link how enhanced acidification or proteolysis impacts on coronavirus fusion.    

(3) Since SARS2 entry into airway epithelial cells expressing ACE2 and TMPRSS2 are most physiologically relevant, the authors should provide data on cell lines accepted to be more physiological mimics of of airway epithelial cells (e.g. Calu-3) or primary airway epithelial cells (even if it is for the key experiments to support the authors main point). Concerns from Reviewers 3 and 4 should be taken seriously in this regard. 

We cannot make any decision about publication until we have seen the revised manuscript and your response to the reviewers' comments. Your revised manuscript is also likely to be sent to reviewers for further evaluation.

Sincerely,

Benhur Lee

Section Editor

PLOS Pathogens

Benhur Lee

Section Editor

PLOS Pathogens

Kasturi Haldar

Editor-in-Chief

PLOS Pathogens

orcid.org/0000-0001-5065-158X

Michael Malim

Editor-in-Chief

PLOS Pathogens

orcid.org/0000-0002-7699-2064

Reviewer's Responses to Questions

**Part I - Summary**

Reviewer #1: Previously, the authors identified an interferon (IFN) inducible gene, NCOA7, that inhibits entry of viruses via the endo-lysosome pathway. SARS-CoV-2 primarily enters host cells by fusing with the plasma membrane wherein transmembrane serine protease TMPRSS2 processes the viral spike protein to facilitate viral entry (1, 2). However, SARS-CoV-2 can also enter cells through the endo-lysosome pathway where cathepsins process the spike protein (2). Using lung epithelial A549 cells expressing the ACE2 receptor, the authors show that in the absence of TMPRSS2, overexpressing NCOA7 decreases SARS-CoV-2 infection. However, in the presence of TMPRSS2, the inhibition mediated by NCOA7 is attenuated. The authors conclude NCOA7 controls SARS-CoV-2 infection only when the virus enters the cell through the endo-lysosomal pathway. It was noted that in comparison to SARS-CoV-2, NCOA7 restriction of infection was not as pronounced with SARS-CoV-1. A major difference between the two viruses is the presence of a polybasic furin cleavage site in SARS-CoV-2 which enables initial processing of the spike protein in the Golgi network during virus production (2). Therefore, the authors investigated the role of the furin cleavage site in NCOA7 mediated restriction. Deletion of the furin cleavage site in SARS-CoV-2 decreased NCOA7 restriction. Conversely, addition of a furin cleavage site to SARS-CoV-1 enhanced NCOA7 restriction of the virus. The authors suggest the furin cleavage site sensitizes SARS-CoV-2 to NCOA7 inhibition within the endo-lysosomal pathway so the virus preferentially exploits TMPRSS2 to enter the host cell at the plasma membrane.

1. V’kovski, P., Kratzel, A., Steiner, S. et al. Coronavirus biology and replication: implications for SARS-CoV-2. Nat Rev Microbiol 19, 155–170 (2021).

2. Peacock, T.P., Goldhill, D.H., Zhou, J. et al. The furin cleavage site in the SARS-CoV-2 spike protein is required for transmission in ferrets. Nat Microbiol 6, 899–909 (2021).

3. Hoffmann, M., Kleine-Weber, H., Pöhlmann, S. A Multibasic Cleavage Site in the Spike Protein of SARS-CoV-2 Is Essential for Infection of Human Lung Cells. Mol Cell 78, 779-784 (2020).

4. Johnson, B.A., Xie, X., Bailey, A.L. et al. Loss of furin cleavage site attenuates SARS-CoV-2 pathogenesis. Nature 591, 293–299 (2021).

5. Lei, X., Dong, X., Ma, R. et al. Activation and evasion of type I interferon responses by SARS-CoV-2. Nat Commun 11, 3810 (2020).

Reviewer #2: In this work, the authors test the role of NCOA7 broadly for restriction of human coronaviruses. Despite its broad efficacy, the overexpression of TMPRSS2 or removal of the spike poly-basic cleavage site can help to avoid this restriction. The authors suggest that viral acquisition of a furin cleavage site and use of the co-receptor may have been selected for to avoid the suppressive effects of interferon. Understanding restriction of coronaviruses is an important topic and this study is well suited for this journal. While the experiments appear to be well-performed from a technical perspective, the lack of statistical analysis makes many of the conclusions drawn by the authors difficult to evaluate. Further, previous work on this restriction factor would allow one to predict with high confidence the results of this study, potentially limiting the impact of this work. My specific comments are below.

Reviewer #3: This is an interesting study of an interferon-stimulated gene and its potential role as a restriction factor for SARS-CoV-2 and other coronaviruses. NCOA7 acts within the endolysosomal pathway and would thus be expected to affect pseudovirus entry only when a) SARS-CoV-2 enters via that route or b) if somehow endolysosomal trafficking alters cell-surface expression of other entry factors. The major weakness of this study is its reliance on the A549 cell line, which is not a preferred model for SARS-CoV-2 infection. The results should be compared with NCOA7 effects on entry in Calu-3 cells or primary human respiratory epithelial cells. One might anticipate that the effects could be attenuated in Calu-3 because of the increased effect of TMPRSS2 in those cells, which the authors note reduces the NCOA7 effects observed.

Broadly, this line of inquiry would be a useful addition to the literature. The major weaknesses of both significance and rigor come from limiting the experiments to a cell type that is not believed to reproduce key properties for SARS-CoV-2 entry.

Reviewer #4: Previous studies have identified two cellular entry pathways for SARS-CoV-2: a mechanism involving endocytosis, cathepsin-mediated cleavage of the viral spike protein (S) and fusion with endosome membranes occurs in many cell lines, whereas cell surface fusion triggered by the serine protease TMPRSS2 has been suggested to be the main mechanism for entry into epithelial cells in organoid models of the respiratory epithelium. The authors of this paper have previously identified an interferon inducible protein NCOA7 that restricts the entry of pH-dependent viruses that infect cells via endocytic routes. NCOA7 interacts with the v-ATPase responsible for the acidification of endocytic organelles and appears to enhance both acidification and the activity of lysosomal hydrolases. In this paper, the authors show that NCOA7 also antagonises SARS-CoV-2 (and other coronaviruses known to infect humans) when virus entry is restricted to the endocytic route. The work is topical and likely to be of interest to not only SARS-CoV-2 researchers, but to virologists and cell biologists in general. However, I do have a few concerns the authors should consider.

Using primarily a respiratory epithelial derived cell line, A549, the authors show that NCOA7 expression inhibits infection by SARS-CoV-2 and by pseudotype viruses expressing SARS-CoV-2 S; they show that interferon-alpha treated cells knocked out for NCOA7 are less well protected than wt cells. They also show that other human coronaviruses and pseudotype viruses expressing S proteins from SARS-CoV-2 variants of concern are similarly affected, and that expression of TMPRSS2 or deletion of the polybasic furin cleavage site reduces NCOA7 mediated inhibition of infection by 2-3 fold. Together, the data make a compelling case for NCOA7 being an interferon inducible antagonist of coronavirus entry via the endocytic route.

**Part II – Major Issues: Key Experiments Required for Acceptance**

Reviewer #1: 1. Mechanistically, it is unclear how the presence of a furin cleavage site enhances NCOA7 restriction of SARS-CoV-2. Acquisition of the furin cleavage site is believed to be directly correlated with increased virus replication pathogenicity (3,4). If the furin cleavage site is negatively affecting the virus as the authors propose, why does the virus maintain this polybasic site?

2. SARS-CoV-2 encodes mechanisms to block and delay IFN production (5). What is the biological significance of IFN inducible NCOA7 in cells that do not express TMPRSS2? Do these cells produce IFN early enough during infection to prevent viral entry?

3. Statistical analyses are lacking in all figures.

Reviewer #2: For all figures, it does not appear that any statistical analysis was ever performed, but in some cases the fold-change is displayed. The appropriate statistical analyses should be applied whenever a comparison is being drawn. Without it, it is almost impossible for a reader to determine if potential changes are meaningful (for example Fig 1G or H).

For many figures, the x-axis describes "Vector volume" to indicate the magnitude of the pseudo typed virus inoculum. While I understand this is convenient, it makes it difficult to compare between viruses. And in some cases, this concern is important evaluate the author's conclusions. For example, Figure 1D and E, we are expected to conclude that NCOA7 restricts SARS-2 but not MLV, however the pseudo virus packaging could have been much more efficient for MLV and a dramatically higher "MOI" could overwhelm restriction. A more standardized quantification such as p24 ELISA would help alleviate this concern.

Figure 3: The cross-experiment comparisons for panels C,D and E,F are not convincing. These comparisons must be done within the same experiment (side-by-side) for a reader to appreciate if the difference between a 4X change in the absence of TMPRSS2 and a 2X change with TMPRSS2 is meaningful or the result of variability between experiments.

SARS-CoV-2 has been demonstrated to not induce strong interferon responses; this seems at odds with the authors suggestion that IFN-induced NCOA7 may have been a strong selection pressure on the virus. This should be discussed.

Reviewer #3: Repeat results in Calu-3 or primary human respiratory epithelial cultures

Reviewer #4: One issue I have with this work is the mechanism of NCOA7 action. To me it seems somewhat counter intuitive that a protein that enhances acidification and lysosomal activity should inhibit entry of a virus that relies on these activities to trigger fusion. The authors argue that excess or premature acidification negatively affects the ability of the virus to undergo fusion. While this may be true for influenza virus, as indicated in their previous paper, the key event for SARS-CoV-2 fusion is proteolytic cleavage of S, and there is no data in this paper to indicate that enhanced acidification/proteolysis impacts on coronavirus fusion.

Secondly, the authors argue that the presence of the furin cleavage site may have allowed the virus to use TMPRSS2 and a cell surface route of entry to evade NCOA7-mediated restriction. It seems to me that the virus without a furin cleavage site can still infect cells by a TMPRSS2 dependent route (Figs 3F and 5), albeit less efficiently. What the furin cleavage site may do is allow more efficient TMPRSS2-mediated activation of S. It’s worth noting that TMPRSS2 expression also reduces NCOA7 inhibition of SARS-1 infection – a virus that lacks the polybasic furin cleavage site? Polybasic cleavage sites do occur in other viruses (e.g. flu), with no obvious impact on the site and underlying mechanism of fusion, as far as I am aware, i.e. for flu at least, there is no evidence the acquisition of a furin cleavage site is linked to escape from NCOA7 mediate restriction.

As entry into respiratory epithelial cells in organoid systems, seems to occur primarily by TMPRSS2-mediated cell surface fusion, does NCOA7 play any role in controlling virus infection and spread in primary tissues, in particular the respiratory epithelium?

**Part III – Minor Issues: Editorial and Data Presentation Modifications**

Reviewer #1: 1. In figure 4, relative light units (RLUs) for control samples are vastly different. For example, RLUs in figure 4B for A549-ACE2 untreated cells without TMPRSS2 are slightly above 103. In figure 4C, the same conditions (A549-ACE2 untreated cells without TMPRSS2) yield 104 RLUs.

2. In figures 1F and 1E, are the cells being treated with IFN� or IFN�? The figure axes are labeled IFN�, but the figure legends state the cells are treated with IFN�.

Reviewer #2: Figure 5E, the text is non-visible in the figure legends, I assume the authors are indicating microliters. Same comment for Sup Fig 1 C,D

Reviewer #3: Biosafety precautions for replication-competent SARS-CoV-2 should be stated explicitly.

"Each experiment shown is representative of three independent experiments." - if this statement were literally true, it would be unacceptable. However, it appears that each experiment was performed in triplicate with the results of each plotted. If such is correct, please alter the statement to reflect this.

Reviewer #4: Given MLV also requires endocytosis for entry, though perhaps not acidification, and that NCOA7 increases lysosomal degradation, is it not a bit surprising that NCOA7 expression has no effect of MLV entry? Or is there a small effect? NB, Rasmussen I, Vilhardt F. describe the entry of MLV as macropinocytosis, not micropinocytosis.

Line 145. Can the authors speculate on why is B.1.351 S protein pseudotypes are less sensitive to NCOA7? Is there any evidence this is giving this VOC an advantage?

Why is there still a 2-3 fold effect of NCOA7 on SARS-2 infection in TMPRSS2 expressing cells? Does NCOA7 expression affect ACE2 and/or TMPRSS2 cell surface expression?

Do BafA, Niclosamide and/or E64D inhibit the effects of NCOA7?

Fig 5. The authors should show that the addition and removal of the polybasic cleavage site does impact on S protein cleavage in virus, or pseudotype, particles.

The authors should cite recent papers from the Haagmans lab.

A number of symbols in the figures need to be corrected (e.g. Fig . 1 F and G, X axis).

PLOS authors have the option to publish the peer review history of their article (what does this mean?). If published, this will include your full peer review and any attached files.

Reviewer #1: **Yes: **Tatyana Golovkina

Reviewer #2: No

Reviewer #3: No

Reviewer #4: No
---

## [Editor Report · Decision Letter 1]

9 Nov 2021

Dear Prof. Malim,

We are pleased to inform you that your manuscript 'TMPRSS2 promotes SARS-CoV-2 evasion from NCOA7-mediated restriction' has been provisionally accepted for publication in PLOS Pathogens.

Best regards,

Benhur Lee

Section Editor

PLOS Pathogens

Benhur Lee

Section Editor

PLOS Pathogens

Kasturi Haldar

Editor-in-Chief

PLOS Pathogens

orcid.org/0000-0001-5065-158X

Michael Malim

Editor-in-Chief

PLOS Pathogens

orcid.org/0000-0002-7699-2064
---

## [Editor Report · Acceptance letter]

17 Nov 2021

Dear Prof. Malim,

We are delighted to inform you that your manuscript, "TMPRSS2 promotes SARS-CoV-2 evasion from NCOA7-mediated restriction," has been formally accepted for publication in PLOS Pathogens.

Best regards,

Kasturi Haldar

Editor-in-Chief

PLOS Pathogens

orcid.org/0000-0001-5065-158X

Michael Malim

Editor-in-Chief

PLOS Pathogens

orcid.org/0000-0002-7699-2064